# Measuring Data Science Automation:
# A Survey of Evaluation Tools for AI Assistants and Agents

**Irene Testini**\*                                                                                   *it370@cam.ac.uk*
*Leverhulme Centre for the Future of Intelligence, University of Cambridge, UK*

**José Hernández-Orallo**                                                                          *jorallo@upv.es*
*Leverhulme Centre for the Future of Intelligence, University of Cambridge, UK*
*Valencian Research Institute for Artificial Intelligence (VRAIN), Universitat Politècnica de València, Spain*

**Lorenzo Pacchiardi**\*                                                                             *lp666@cam.ac.uk*
*Leverhulme Centre for the Future of Intelligence, University of Cambridge, UK*

**Reviewed on OpenReview:** *https://openreview.net/forum?id=MBOTCLfLn1*

## Abstract

Data science aims to extract insights from data to support decision-making processes. Recently, Large Language Models (LLMs) have been increasingly used as *assistants* for data science, by suggesting ideas, techniques and small code snippets, or for the interpretation of results and reporting. Proper automation of some data-science activities is now promised by the rise of LLM *agents*, i.e., AI systems powered by an LLM equipped with additional affordances—such as code execution and knowledge bases—that can perform self-directed actions and interact with digital environments. In this paper, we survey the evaluation of LLM assistants and agents for data science. We find (1) a dominant focus on a small subset of goal-oriented activities, largely ignoring data management and exploratory activities; (2) a concentration on pure assistance or fully autonomous agents, without considering intermediate levels of human-AI collaboration; and (3) an emphasis on human substitution, therefore neglecting the possibility of higher levels of automation thanks to task transformation.

## 1 Introduction

Large Language Models (LLMs) (Brown et al., 2020) and their multimodal extensions first caught public prominence by powering capable chatbots that are now widely used as *assistants* to humans in several tasks, such as summarising documents (Liu et al., 2023c), performing translations (Zhu et al., 2023), and creating code snippets (Guo et al., 2023). These LLM assistants take instructions from a human in the form of a prompt and return an answer, with the human retaining control over planning and decision-making by determining the sequence of actions to follow and deciding how much to rely on the assistant's output; while they can use tools, their usage is generally prescribed at certain steps. Now, attention is increasingly dedicated to "LLM *agents*" (Wang et al., 2024) that can autonomously and iteratively decide a sequence of actions to take and repeatedly interact with an external (digital) environment, being equipped with affordances and tools such as code execution (Huang et al., 2024b), internet access (Zhou et al., 2024), knowledge bases (Chen et al., 2024), and operating system control (Liu et al., 2023a), which the agents can decide to use of autonomously.

In this paper we focus on how assistants and agents for data science applications *are being evaluated*. Our aim is to provide an overview of current evaluation set ups, and highlight the gaps in scopes of evaluation, rather than reviewing the performance of the models themselves. Data science is the process of handling and analysing data to extract insights that support decision-making in science, business, or other contexts.

---

\*Equal contribution.

Data science may deal with different modalities of data (tabular, images, audio, etc.) but it always involves writing and interpreting information represented as text, whether in the form of code or natural language, and processing images, such as to present information or models. This dominance of textual modalities, combined with the vast amount of relevant online material that LLMs leverage during both training and inference, makes data science well-suited for automation by LLMs. Indeed, since the early days of LLMs (Chandel et al., 2022), LLM assistants and agents have been used in data science applications. In this work, we survey evaluation tools measuring performance on tasks across the data science pipeline, offering, to our knowledge, the first comprehensive review of LLM evaluation for core data science tasks[1].

Data science is highly multidisciplinary and involves a breadth of activities, combining fields such as statistics, machine learning, and data engineering with tasks such as understanding business needs, writing reports, and preliminary research. Therefore, to compare the data science automation evaluation tools in our survey, we adopt the data science task taxonomy of Martínez-Plumed et al. (2019), which expands the traditional CRISP-DM framework to include activities related to data management and exploratory analysis (see Fig. 1, reproducing Fig. 3 from Martínez-Plumed et al. 2019), and classify each evaluation tool by the activities it requires subjects to perform and explicitly evaluates. Moreover, in our survey we specifically consider the level of autonomy each evaluation targets—whether the subjects are agents, assistants, or intermediate forms, such as LLM agents operating under close human supervision that correct their actions as needed. Further, we also analyse the way in which the tasks are framed and evaluated, to understand if they measure the ability of AI systems to simply *substitute* humans or if instead they consider that the AI system can *transform* the task in deeper ways, such as by bringing functional improvements (for instance, LLMs may not need to produce visualisations to perform data exploration; see Sec. 2.1).

It is worth stressing how, in our work, we do *not* overview developments in state-of-the-art LLM assistants or agents for data science (referring interested readers to other works, Sun et al., 2024) nor analyse the current performance of such systems across the various data science activities and autonomy levels. While such efforts would be valuable and may reveal strong correlations between LLM systems' performance on specific tasks, our work addresses a more foundational need: to establish taxonomies across which data science evaluations can be structured and to identify gaps in the evaluation landscape that must be filled before a comprehensive evaluation of LLM abilities can be carried out. Therefore, by focusing on activities and autonomy, we make the following findings:

- Most evaluation works individually target a (small) subset of data science activities (Tables 2 and 3); a few works cover multiple activities (Secs. 4.2 and 4.3), but none cover all of them. Taken together, the surveyed works cover the landscape of data-science activities in a biased fashion, chiefly over-representing "goal-oriented" activities, such as data preprocessing, producing plots in specified formats, or building predictive models for predetermined targets. This prevents the identification of correlations between AI systems' performance on different activities. Only a few studies (Cheng et al., 2023; Sahu et al., 2025; Majumder et al., 2024) give prominence to open-ended, exploratory aspects of data science (such as interpreting client needs within a business context, creatively exploring datasets and proposing potential uses) or data management(Yu et al., 2018; 2019b;a; Lei et al., 2024).

- Most studies focus either on assistants following human-defined actions or fully autonomous agents, overlooking more realistic scenarios of intermediate LLM–human collaboration (also referred to as "centaur evaluations", Haupt & Brynjolfsson, 2025). Exceptions include Li et al. (2025c;b;a) (agents with simulated human users) and Yu et al. (2019a) (assistants aiding users in clarifying data tasks). Due to this near-binary focus, we organise the surveyed works according to the focus on assistants (Sec. 3) and agents (Sec. 4), highlighting when they assess intermediate autonomy.

- More than half of the evaluations we survey implicitly assume that AI will substitute humans without functionally changing the tasks, either in assuming the steps by which a task is solved are the same a human would follow (Yu et al., 2019a; Zhang et al., 2024b; Chen et al., 2024) or by scoring the task referring to human-produced output, despite there not being a single ground

---

[1] While many studies assess LLMs on general coding tasks (Jimenez et al., 2024) and planning (Valmeekam et al., 2023), our survey focuses specifically on evaluations within the data science domain. This includes works that target data science-specific coding as well as other data science activities that are often overlooked in existing evaluations.

truth (Song et al., 2025; Huang et al., 2024b; Hu et al., 2024; Jing et al., 2024; Chen et al., 2024). Examples of works also rewards agents that functionally transform the task to solve it by scoring the final output are Pietruszka et al. (2024); Cheng et al. (2023); Li et al. (2025b); Chan et al. (2024); Lei et al. (2024); Majumder et al. (2024); Sahu et al. (2025).

The paper is structured as follows: Sec. 2 discusses some fundamental concepts on technological transformation, assistance and autonomy, LLM evaluation, and data science automation and its activities. Secs. 3 and 4 dive deep into works evaluating LLM assistants and agents, respectively. Sec. 5 summarises the challenges in data science evaluation and suggests future directions for more effective and comprehensive evaluation.

## 2 Background and related work

### 2.1 Levels of technological transformation

A naive perception of how technology transforms processes and activities is automation by *substitution*: a human performing a task is replaced by a machine without functionally transforming the task. To evaluate this, a sample of tasks representing how humans tackle an activity is collected and the machine is tested on them (Eriksson et al., 2025). The Substitution-Augmentation-Modification-Redefinition (SAMR) model (Puentedura, 2006; Hamilton et al., 2016) identifies substitution as the lowest level of transformation and outlines subsequent levels with progressively greater degrees of transformation: augmentation, where the machine substitutes the human with some functional improvement; modification, where the task is significantly redesigned to allow automation; redefinition, in which the whole activity is redesigned, even creating new tasks. Much of the debate around AI-powered automation focuses on the two bottom levels (augmentation and substitution), but the real penetration of AI technology is happening at the top levels of modification and redefinition (Brynjolfsson, 2022; Brynjolfsson et al., 2025), which hold the potential to achieve higher levels of automation. Indeed, we did not "automate away the jobs of lamplighters by building robots capable of carrying ladders and climbing lampposts" (Frey & Osborne, 2023). Importantly, activities that have already been substituted can be iteratively transformed further as technology improves. Evaluating progress is therefore much more complex than if substitution was the only force at play: a robot substituting human lamplighters in carrying and climbing ladders would have scored highly in turning on gas lamps, but the redefinition afforded by electricity led to automating street lights, rendering robotic lamplighter needless. Similar considerations apply when evaluating AI progress in automating complex processes composed of many activities, such as data science: for instance, LLMs may not need to produce high quality visualisations to perform successful data exploration, as they may be able to directly interpret large tables of data. To effectively evaluate modification and redefinition, AI evaluation should allow AI systems to perform activities differently from humans by rewarding the achievement of broad objectives.

### 2.2 Assistance and autonomy

Related yet orthogonal to the SAMR model is the distinction between assistance and autonomy (Shneiderman, 2020): in an assistive situation, a human uses the technology while retaining control of the process and only having some well-defined parts automated or improved. Assisted driving or writing are good examples: the process becomes more efficient and safe because of the use of technology. On the contrary, in an autonomous situation, the technology performs the task independently and has more freedom to choose how. Of course, autonomy exists on a spectrum: intermediate levels include, for example, technology operating independently while a human oversees the sequence of steps and retains the ability to halt operations. In relation to this, Cihon et al. (2024) defined levels of agent features relevant to autonomy. Their classification assigns high autonomy to agents acting fully autonomously, whereas the intermediate and lower levels correspond to agents consulting humans either at termination or at each step. This aligns with our understanding of autonomy levels, which additionally includes an even lower level where a human assigns a specific task to an assistant. Hence, holistic AI evaluation should take into account quality of the result and the level of human labour, which limits the impact of the technology in the long term. Importantly, for all levels of autonomy, the technology can perform the task in a way that places the automation at any level of the SAMR hierarchy.

### 2.3 LLM evaluation

The area of AI evaluation (Burden et al., 2025) mostly relies on tasks encapsulated in input-output benchmarks with a reference output for each example. For LLMs, these input–output pairs are most often Q&A examples used to evaluate assistants (Chang et al., 2023; Guo et al., 2023) or autonomously acting agents (Wang et al., 2024; Yehudai et al., 2025). While the use of natural language gives a perspective of breadth, recent works highlighted how current evaluation practice fails to measure realistic human-LLM interaction (Haupt & Brynjolfsson, 2025) and real-world impact (Burden et al., 2025; Reiter, 2025). This agrees with our findings that evaluations for data science mostly fail to capture intermediate levels of LLM-human collaboration and concentrate on evaluating substitution rather than higher levels of transformation (Sec. 2.1). Indeed, Chang et al. (2023) and Haupt & Brynjolfsson (2025) highlighted human-in-the-loop testing and evaluations in an open environment as important directions, and Wang et al. (2024) identified a shift towards end-to-end tasks requiring human evaluators and versatile metrics, yet most evaluations today only consider a subset of tasks in the data science pipeline, with a few exceptions (Sec. 4.3).

A few studies evaluate truly long-horizon scenarios or quantify the human effort they still require. Wang et al. (2023) and Park et al. (2023) showed that agents can sustain hours-to-days of open-ended play or social simulation, but both exposed failure modes that need periodic human nudges. Quantitatively, Liu et al. (2023a) found that commercial models needed a median of 2.4 human corrections per task on a general agent benchmark, whereas open-source models needed 5–8. Recently, Kwa et al. (2025) showed that autonomous agents are progressively conquering tasks that take humans longer to complete when considering a fixed success rate (e.g., 50%), but performance still progressively degrades on tasks requiring more than 10 seconds. Recently, Kwa et al. (2025) stratified tasks according to the mean execution time humans require to complete them, then measured AI model performance on each subset of tasks within specific time ranges (e.g., tasks taking around 10 minutes for humans). Plotting human-estimated task completion time against AI success rate, they demonstrated that AI success rates decrease as human completion time increases. However, when examining the human time range at which different AI models achieve a 50% success rate, more recent models consistently reach this threshold on tasks requiring longer human completion times, indicating progressive improvement in tackling more complex, time-intensive tasks.

### 2.4 Data science automation

Automating data science was a topic of research even before LLMs became commonplace. Bie et al. (2021) argued that the technical and domain knowledge required to solve data science tasks motivated efforts toward automation. The authors categorised data science tasks into four main quadrants, defined by two axes—degree of open-endedness and dependence on domain context—highlighting that model-building activities are more easily automated (e.g., through AutoML approaches, Hutter et al. 2019; Gijsbers et al. 2024) due to their lower open-endedness and context dependence. They also identified three forms of automation: mechanisation, composition, and assistance. Assistance corresponds to our interpretation of the term while mechanisation and composition can be grouped under our umbrella of automation (Sec. 2.2), but differ in focusing respectively on small parts of the process or on the overall pipeline; in our work, we do not make this distinction and instead rely on the activities of Martínez-Plumed et al. (2019) to identify how many elements of the pipeline each evaluation work covers. After Bie et al. (2021) published their survey, numerous works built LLM agents to automate data science. Their evolution, capabilities, and applications across the data science pipeline are reviewed by Sun et al. (2024); the authors, however, did not address LLM *evaluation* for data science, which is the focus of our work. More recently, (Hu et al., 2025) proposed a taxonomy for the data ecosystem: Data Management, which includes data collection, data storage, and data preprocessing; Data Analysis, which includes model evaluation, data interpretation, and decision making; and Data Visualisation. This taxonomy partly overlaps with that in Martínez-Plumed et al. (2019) which we use in our work (Sec. 2.5), but misses some of the most exploratory aspects. Finally, Chintakunta et al. (2025) conduct a systematic mapping study examining the application of LLMs in data science. Considering a 5-stage decomposition of data science, they find that most papers apply LLMs in Data Exploration and Analysis, followed by Model Building and Evaluation, and Data Collection and Preparation, with Deployment and Problem Definition unexplored – which mostly matches our findings, obtained with a finer task decomposition. Moreover, from the surveyed papers, a consensus on research gaps emerges, including the

need to expand to complex, real-world evaluation tasks and to enhance human-AI interaction by creating more user-friendly interfaces; these echo our findings on the lack of human-AI interaction evaluations and the artificial nature of many evaluation tools, for instance by the adoption of a "substitution" perspective. In addition, our survey complements Chintakunta et al. (2025) by providing a deeper analysis of evaluation.

## 2.5 The activities of data science

Table 1: Data-science activities and brief definition (complete definitions in Appendix A).

| Activity (abbr.) | Brief definition |
| --- | --- |
| *Goal-oriented (CRISP-DM)* | |
| Business Understanding (BU) | Define the problem and draft a plan that meets business requirements |
| Data Understanding (DU) | Collect and explore data to spot useful subsets, insights, or issues |
| Data Preparation (DP) | Build the final analysis dataset via selection, cleaning, and transformation |
| Modelling (M) | Apply modelling techniques, tune their parameters and evaluate models |
| Evaluation (E) | Check that the business objectives are met, with no overlooked issues |
| Deployment (Dep) | Deliver the model's outputs in a usable form (report, integration, etc.) |
| *Exploratory* | |
| Goal Exploration (GE) | Identify business goals that could be addressed with data |
| Data Source Exploration (DSE) | Discover new, valuable data sources |
| Data Value Exploration (DVE) | Judge the potential value that can be extracted from the data |
| Result Exploration (RE) | Connect data-science results back to business goals |
| Narrative Exploration (NE) | Craft meaningful (visual or textual) stories from the data |
| Product Exploration (PE) | Devise services or applications that turn extracted value into products |
| *Data-management* | |
| Data Acquisition (Acq) | Obtain or generate relevant data (e.g., via sensors or apps) |
| Data Simulation (Sim) | Simulate systems to generate data and explore causal "what-if" scenarios |
| Data Architecting (Arch) | Design the logical/physical layout and integration of data sources |
| Data Release (Rel) | Make data accessible through databases, APIs, or visualisations |

Many taxonomies of data science activities exist (see Martínez-Plumed et al., 2019, Sec 2). One of the most popular is CRISP-DM (Cross Industry Standard Process for Data Mining, Chapman, 2000), which considers projects as *goal-oriented*, with a pre-defined objective that can be approached by "mining" data through an approximately sequential process, from problem framing to solution delivery. However, Martínez-Plumed et al. (2019) argues that this goal-oriented, pre-collected-data perspective ignores many tasks of modern data science, where exploration is essential and the data takes centre stage rather than serving as a fixed backdrop. Consequently, Martínez-Plumed et al. (2019) expands this taxonomy by proposing a sequence of *exploratory* activities that underscore the less prescriptive nature of data science, re-framing it as an investigative endeavour; and *data management* activities that do not assume data is already given and require to fetch more data from different sources. We provide a list of the activities introduced in Martínez-Plumed et al. (2019) and a concise definition in Table 1. Note that not all modern data-science projects include every activity, nor is the order of activities fixed as in the CRISP-DM framework. Instead, each project follows its own "trajectory" in the space of data-science tasks (Martínez-Plumed et al., 2019). Moreover, the distinction between activities may be blurred.

# 3 Evaluating LLM assistants in data science

In this section, we focus on evaluations of LLMs as assistants, namely prompting them in a fixed, pre-determined manner without letting them independently determine the sequence of steps. Table 2 shows the

surveyed papers and the activities (Sec. 2.5) they cover; a double tick marks an activity that is explicitly assessed, whereas a single tick marks an activity that is required for completing the tasks but not directly assessed.

Table 2: Data science activities covered by the surveyed LLM assistants evaluation works. See Sec. 2.5 for definition of the acronyms. A double tick refers to an activity explicitly evaluated, while a single tick refers to an activity necessary for succeeding in the tasks but not explicitly evaluated.

| Papers | Goal-oriented | | | | | | Exploratory | | | | | | Data Management | | | |
|---|---|---|---|---|---|---|---|---|---|---|---|---|---|---|---|---|
| | BU | DU | DP | M | E | Dep | GE | DSE | DVE | RE | NE | PE | Acq | Sim | Arch | Rel |
| ARCADE (Yin et al., 2022) | - | ✓ | ✓✓ | - | - | - | - | - | ✓ | - | - | - | - | - | - | - |
| AssistedDS (Luo et al., 2025) | - | ✓ | ✓ | ✓ | - | - | - | - | ✓✓ | - | - | - | - | - | - | - |
| CERT (Zan et al., 2022) | - | - | ✓✓ | - | - | - | - | ✓ | ✓ | - | - | - | - | - | - | - |
| CoSQL (Yu et al., 2019a) | - | ✓ | - | - | - | ✓✓ | ✓✓ | - | - | - | - | - | - | - | ✓✓ | - |
| DS-1000 (Lai et al., 2023) | - | ✓ | ✓✓ | ✓✓ | - | - | - | - | - | - | - | - | - | - | - | - |
| DS-Bench (Ouyang et al., 2025) | - | ✓ | ✓✓ | ✓✓ | - | - | - | - | - | - | - | - | - | - | - | - |
| DSP (Chandel et al., 2022) | - | ✓ | ✓✓ | - | - | - | - | - | - | - | - | - | - | - | - | - |
| FeatEng (Pietruszka et al., 2024) | - | ✓✓ | ✓✓ | ✓ | - | - | - | ✓ | ✓ | - | - | - | - | - | - | - |
| GPT4-DA (Cheng et al., 2023) | ✓ | ✓✓ | ✓ | ✓✓ | - | ✓✓ | - | - | ✓✓ | ✓✓ | ✓✓ | - | - | - | - | - |
| HardML (Pricope, 2025) | - | ✓ | ✓ | ✓ | - | - | - | - | ✓ | - | ✓ | - | - | - | - | - |
| LIDA (Dibia, 2023) | - | ✓✓ | ✓✓ | - | ✓ | - | ✓✓ | ✓ | ✓✓ | - | ✓✓ | - | - | - | ✓ | ✓✓ |
| SParC (Yu et al., 2019b) | - | ✓ | - | - | - | - | - | - | - | - | - | - | - | - | ✓✓ | - |
| Spider (Yu et al., 2018) | - | ✓ | - | - | - | - | - | - | - | - | - | - | - | - | ✓✓ | - |
| Spider 2.0-Lite (Lei et al., 2024) | - | ✓ | - | - | - | - | - | - | - | - | - | - | - | - | ✓✓ | - |
| Spider 2.0-Snow (Lei et al., 2024) | - | ✓ | - | - | - | - | - | - | - | - | - | - | - | - | ✓✓ | - |
| StatLLM (Song et al., 2025) | - | ✓ | ✓✓ | ✓✓ | ✓ | - | - | - | - | - | - | - | - | - | - | - |

First, many works evaluate LLMs used to generate code for specific steps of data science, such as preprocessing data given a template, fixing bugs, or producing visualisations given instructions or prerequisites. In particular, **ARCADE** (Yin et al., 2022) and **CERT** (Zan et al., 2022) focus on Data Preparation and related activities with specific Python libraries. ARCADE is a benchmark consisting of 1,082 coding problems involving data wrangling and Exploratory Data Analysis (EDA), defined as Jupyter notebooks, that require Python's `Pandas` library; for example, a problem could involve extracting min and max values from a dataframe and answer questions such as "In which year was the most played game added?". CERT instead introduces two benchmarks (PandasEval and NumpyEval), each consisting of 101 tasks manually reworked for coherence and consistency from StackOverflow[2] problems tagged as relevant to `Pandas` and `NumPy` respectively; for problems whose solution is a function, 20 test cases are included, while the correctness of the predicted variable is checked for the other problems. Relatedly, **DSP** (Chandel et al., 2022) contains problems instantiated in 306 pedagogical Jupyter notebooks with 92 associated datasets, covering data manipulation, cleaning, and wrangling (parts of Data Preparation). Similarly to CERT, the correctness of the task is automatically graded with test cases. An example problem would be "Show the correlation between population density in 2023 and 2050, rounded to 2 decimals". Similarly, **DS-1000** (Lai et al., 2023) consists of 1,000 coding problems extracted from StackOverflow, spanning Python libraries such as `NumPy`, `Pandas`, `SciKit-Learn`, `TensorFlow`, `matplotlib`, `SciPy`, and `PyTorch`. The problems are manually perturbed to circumvent the issue of memorisation in LLMs and cover Data Preparation and Modelling. The problems are scored through multi-criteria execution-based evaluation metrics that rely on test cases and constraints to check whether the output relies on specific packages and functions. See Fig. 2 for an example problem and evaluation set-up. Ouyang et al. (2025) extend DS-1000 to create **DS-Bench** by adding `Seaborn`, `Keras` and `LightGBM`. To build the benchmark, they first define a broad task scope and collect Python seed code from GitHub using DS-1000's reference code and corresponding StackOverflow answers. An automated LLM pipeline then transforms each snippet to avoid memorisation. Candidates are filtered by properties (compilability, stars, API calls) and functionality (must pass at least one LLM-generated test). For each

---

[2] https://stackoverflow.com/questions

surviving candidate, an LLM generates 200 test cases and a problem description—complete with an introduction, function signature, input/output formats and examples. After manual review, 1,000 problems are selected. Performance is measured with pass@$k$: a problem is solved if any $k$ generated samples pass the unit tests. Instead, **FeatEng** (Pietruszka et al., 2024), evaluates LLMs' ability to produce a Python function for engineering data features (thus addressing Data Understanding and Preparation) suitable for downstream modelling tasks. The authors select datasets based on their popularity on Kaggle and ensuring broad domain coverage, reaching a total of 101 tasks. Notably, and in contrast to the older works described above, where questions admitted fixed ground truths, performance is measured in terms of the reduction in error of a model trained on the extracted features compared to a baseline model trained on the original, untransformed data. Therefore, this allows AI models to go beyond simply substituting humans and reach higher levels of task transformation (Sec. 2.1). **StatLLM** (Song et al., 2025) instead focuses on statistical Modelling, assessing LLMs' ability to generate code to solve a dataset of 207 statistical analysis tasks assembled from various public online resources, including descriptive statistics, hypothesis testing, regression and ANOVA, generalised linear models, survival analysis, model selection, and non-parametric statistics; tasks might require the LLM to run a specific model on a variable in a given dataset, or to plot a variable. Uniquely, the LLM has to generate SAS code; evaluation is carried out using Natural Language Processing (NLP) metrics to compare the generated code against a human gold standard, thus being grounded in *substitution* (Sec. 2.1).

Considering Data Management activities, **Spider** (Yu et al., 2018), **SParC** (Yu et al., 2019b) and **CoSQL** (Yu et al., 2019a) (all from the same research group) evaluate conversational database querying systems translating natural language into SQL queries (part of Data Architecting but also requiring Data Understanding). These works build on the same 200 databases from 138 domains: Spider consists of 10,181 manually crafted questions and 5,693 unique SQL reference queries and evaluates the generated queries with matching of SQL components or the overall query to the reference one, or with the accuracy of the execution. SParC expands Spider, which contains only single-turn questions, by simulating multi-turn interactions and therefore introducing context dependence: annotators chained Spider tasks together in a conversational flow resulting in 4,298 question sequences with 12,726 questions. Performance is evaluated in terms of exact set match (per turn), and interaction match (full sequence accuracy); however, this does *not* evaluate the ability of the AI system to interact with a user successfully. This is done in CoSQL, which also includes task where the system must identify ambiguous questions needing clarifications and unanswerable queries (accuracy is evaluated using dialogue act labels). This makes CoSQL unique in addressing intermediate levels of automation for assistants (Sec. 2.2). The clarifications are then included in the context the system uses to determine the correct SQL query, scored using exact match or component match. However, despite being inserted in the context of a conversation, only one system answer at a time is evaluated, therefore still considering the paradigm of *substitution* (Sec. 2.1). Natural language summaries of the query output produced by the system are also evaluated (with the BLEU score). Overall, CoSQL comprises over 30,000 dialogue turns and 10,000 annotated SQL queries, derived from approximately 3,000 dialogues collected by having users interact with a mock interface controlled by an expert and simulating real-world database query scenarios. Finally, the same authors recently introduced (Lei et al., 2024), **Spider 2.0-Lite**, consisting of 547 test instructions mapped to 158 real databases hosted on BigQuery, Snowflake and SQLite and solely scored based on execution accuracy, and **Spider 2.0-Snow**, re-hosting the same 547 questions on Snowflake to spotlight one dialect while keeping identical self-contained evaluation.

Moving to the exploratory aspects of data science, **LIDA** (Dibia, 2023) introduces a system generating data visualisation and infographics by prompting LLMs in a structured manner to provide a summary of the dataset (Data Understanding), formulate data exploration goals (Data Value Exploration), generate code specifications for the visualisations (Goal Exploration), and generate stylised graphics based on the previous output (Narrative Exploration). This also covers aspects of Data Release as it involves making data accessible through visualisations. The system is accompanied by an evaluation tool, based on 57 datasets sourced from the `Vega` datasets[3] repository; two metrics are used: visualisation error rate, computed as the percentage of generated visualisations that result in code compilation errors; and visualisation quality, in which GPT-4 (Achiam et al., 2023) is tasked with assessing the quality of the generated visualisations across 6 dimensions: code accuracy, data transformation, goal compliance, visualisation type, data encoding, and

---

[3]https://github.com/vega/vega-datasets

aesthetics. Instead, **AssistedDS** (Luo et al. (2025)) is a benchmark designed to evaluate how well LLMs leverage domain knowledge to improve predictive performance on tabular datasets. It tests the model's ability to critically assess, filter, and apply both helpful and harmful external information, using synthetic datasets with controlled feature-label relationships and Kaggle datasets augmented with high- and low-rated notebook insights. Evaluation focuses on code quality and predictive accuracy across, and the same task is evaluated by providing different configurations of domain knowledge. By measuring how effectively models identify and apply valuable domain insights, AssistedDS directly targets the core of Data Value Exploration.

While the above works focus on single steps of the data science pipeline, Cheng et al. (2023) evaluates GPT-4 as a data analyst on end-to-end data mining problems (excluding several exploratory steps and the entirety of data management). In particular, they provide GPT-4 with a database schema (Data Understanding) and a real-world business question (Business Understanding) and tasks it with extracting the relevant data (Data Preparation, Data Value Exploration), conducting Modelling, generating visualisations and producing an analysis (Deployment, Narrative Exploration, Result Exploration). GPT-4 is embedded within a framework (referred to as **GPT-4DA**), in which it is first prompted to generate code that is executed to produce graphs and a text file containing the generated data, and then prompted again to generate an analysis comprising five insights derived from the textual data (excluding the figures). They devise three evaluation metrics for the generated figures (correctness of data and information, chart type, and aesthetic), and four evaluation metrics for the generated insights (correctness of data and information, alignment with question, complexity, and fluency). By using these broad metrics, GPT-4 is free to solve the task in ways different from what humans would do, thus reaching higher levels of transformation (Sec. 2.1). They test this pipeline on the NvBench dataset (Luo et al., 2021) and employ six human professionals to evaluate GPT-4 (using a rubric detailing the above metrics) and a professional (human) data analyst as baseline. While involving humans leads to more comprehensive understanding of performance, it also makes running the evaluation more costly and less reproducible.

Data science involves additional skills other than coding. For example, domain knowledge in data science is essential. To evaluate this, Pricope (2025) introduce **HardML**, a benchmark of 100 multiple-choice questions, designed to challenge experienced data science professionals, assessing advanced reasoning skills and domain knowledge. The questions are original, handcrafted, and may include multiple correct answers; they span various topics such as natural language processing, computer vision, statistics and statistical modelling, classical machine learning, and cover activities such as Data Understanding and Preparation, Modelling, Data Value Exploration and Narrative Exploration. An example question is: "An AI company just shipped a new foundational language model. They claim they have trained it for 2.79M H800 hours on 14.8T tokens. Upon further research, looking at Nvidia card specs, you find 3,026 TFLOPs/s of FP8 performance with sparsity, or typically half this (1.513e15 FLOPs/s) without sparsity. Moreover, you find out that they used FP8 FLOPs without structured sparsity. Given that the model has 37B activated parameters, roughly what hardware utilization did they achieve? Select the closest." Importantly, whilst several activities are evaluated by the benchmark, each question only targets a single activity; moreover, the majority of the questions focus on reasoning capabilities and coding for machine learning and various aspects of deep learning engineering.

From this overview, it is evident how nearly all LLM assistant evaluation works focus on code generation, and how there is a concentration on the goal-oriented activities of Data Understanding, Data Preparation and Modelling (Table 2), with a paucity of evaluation works for exploratory and, even more, data management activities, except for Data Value Understanding, and a few works touching on Narrative Exploration, Goal Exploration, Data Source Exploration, Result Exploration, Data Architecting, and Data Release.

# 4 Evaluating LLM agents in data science

In this section, we consider evaluations of LLM agents, which augment LLMs with a set of affordances and allow them to determine the sequence of steps they go through by iterative prompting. We also consider works evaluating agents with (simulated) user interaction. Table 3 shows the papers we overview and the activities they cover (Sec. 2.5); a double tick marks an activity that is explicitly assessed, whereas a single tick marks one that is vital for completing the tasks but not directly assessed.

Table 3: Data science activities covered by the surveyed LLM agent evaluation works. See Sec. 2.5 for definition of the acronyms. A double tick refers to an activity explicitly evaluated, while a single tick refers to an activity necessary for succeeding in the tasks but not explicitly evaluated.

| Papers | Goal-oriented | | | | | | Exploratory | | | | | | Data Management | | | |
|---|---|---|---|---|---|---|---|---|---|---|---|---|---|---|---|---|
| | BU | DU | DP | M | E | Dep | GE | DSE | DVE | RE | NE | PE | Acq | Sim | Arch | Rel |
| BLADE (Gu et al., 2024) | - | ✓✓ | ✓✓ | ✓✓ | - | ✓ | - | - | ✓ | - | ✓✓ | - | - | - | - | ✓ |
| BiasBenchmark (Li et al., 2025b) | - | ✓✓ | - | - | - | - | - | - | - | - | - | - | - | - | - | - |
| CoTa (Li et al., 2025c) | - | ✓ | ✓✓ | ✓✓ | - | ✓ | ✓ | - | - | - | ✓ | - | - | - | - | - |
| CSR-Bench (Xiao et al., 2025) | - | - | ✓ | ✓ | ✓ | ✓✓ | - | - | - | - | - | - | - | - | - | - |
| Data-Copilot (Zhang et al., 2024a) | - | ✓ | ✓✓ | ✓ | - | - | - | - | ✓ | - | - | - | - | - | - | - |
| DA-Code (Huang et al., 2024b) | - | ✓✓ | ✓✓ | ✓✓ | ✓ | - | - | - | ✓ | - | - | - | - | - | - | - |
| DiscoveryBench (Majumder et al., 2024) | ✓ | ✓ | ✓ | ✓ | - | ✓ | ✓✓ | ✓ | ✓ | - | ✓✓ | - | - | - | ✓ | - |
| DSBench (Jing et al., 2024) | - | ✓ | ✓ | ✓✓ | ✓ | ✓ | ✓ | - | ✓ | ✓ | ✓ | - | - | - | - | - |
| DS-Eval (Zhang et al., 2024b) | ✓ | ✓✓ | ✓✓ | ✓✓ | - | ✓✓ | - | - | - | - | ✓✓ | - | - | - | - | - |
| IDA-Bench (Li et al., 2025a) | - | ✓ | ✓ | ✓✓ | - | ✓✓ | - | - | - | - | - | - | - | - | - | - |
| InfiAgent-DABench (Hu et al., 2024) | - | ✓ | ✓ | ✓✓ | - | - | - | - | ✓ | - | ✓ | - | - | - | - | - |
| InsightBench (Sahu et al., 2025) | ✓ | ✓ | ✓ | ✓ | ✓ | ✓✓ | ✓ | - | ✓ | ✓✓ | ✓✓ | - | - | - | - | ✓✓ |
| MLAgentBench (Huang et al., 2024a) | - | ✓ | - | ✓✓ | ✓ | ✓ | - | ✓ | - | ✓ | ✓ | - | - | - | - | - |
| MLE-Bench (Chan et al., 2024) | - | ✓ | ✓ | ✓✓ | - | ✓ | - | - | ✓ | - | ✓ | - | - | - | - | - |
| MLGym (Nathani et al., 2025) | - | ✓ | ✓ | ✓✓ | ✓ | - | - | - | ✓ | - | - | - | - | - | - | - |
| RE-Bench (Wijk et al., 2024) | - | - | - | ✓✓ | - | - | - | - | - | - | - | - | - | - | - | - |
| ScienceAgentBench (Chen et al., 2024) | - | ✓✓ | ✓✓ | ✓✓ | - | ✓✓ | ✓ | - | ✓ | - | ✓✓ | - | - | - | - | - |
| Spider 2.0 (Lei et al., 2024) | - | ✓✓ | - | - | - | - | ✓✓ | - | - | - | - | - | - | - | ✓✓ | - |
| SUPER (Bogin et al., 2024) | - | - | - | ✓ | ✓ | ✓✓ | - | - | - | - | - | - | - | - | - | - |
| WebDS (Hsu et al., 2025) | ✓ | ✓ | ✓✓ | ✓ | - | ✓✓ | - | ✓ | - | ✓✓ | ✓✓ | - | - | - | - | ✓ |

## 4.1 Works targeting specific goal-oriented activities

Many works aim to evaluate LLM agents on individual goal-oriented activities of the data science pipeline. Starting from Data Understanding, Li et al. (2025b) introduce **BiasBenchmark**, which evaluates the ability to detect biases in datasets. To build the benchmark, they select 5 datasets from prior bias mitigation research and 100 demographic-related features or their combinations and craft (using an LLM playing the role of a user) possible bias detection queries, including intentionally ambiguous questions. Then, during evaluation, a bias detection query is passed to the evaluated LLM agent to test whether it is able to effectively detect if the bias exists by analysing the provided data; clarification questions posed by the agent are automatically answered by the LLM-based user simulator according to the original task specifications, ensuring consistency and reproducibility. The agent must quantify the bias level according to a 5-level scale, which is then compared to the ground truth obtained by measuring five widely-used bias detection metrics. However, scoring the final bias level allows to agent to determine it in potentially novel ways, therefore going above human substitution (Sec. 2.1). They also evaluate the agent's intermediate process, by developing an agent-based automated evaluation system that looks at the evaluated agent's logs and produces performance rating levels for five aspects: user communication, task planning, tool invocation, dynamic plan adjustment and result analysis. Interestingly, the first one (user communication) scores the ability of the agent to ask clarification questions to the (simulated) user who set the task.

Instead, **Data-Copilot** (Zhang et al., 2024a) introduces an LLM agent for data wrangling that, given a dataset schema, independently explores potential user requests and generates modular code to address them, which is then leveraged in the deployment stage. To benchmark it, the authors release 547 test requests drawn from 173 human seeds plus a larger 3547-request self-exploration pool. The tasks rely on financial data and touch upon Data Value Exploration, Data Understanding and Preparation, and Modelling. Each test case is accompanied by a human-curated answer table and they are jointly (manually) annotated with four labels for dataset analysis: task difficulty, request rationality, expression ambiguity, answer accuracy. System performance is measured with GPT-4-based Pass@1 scoring against the gold tables (plus an image check) and with the number of tokens used.

Moving towards Modelling, **MLE-Bench** (Chan et al., 2024) and **RE-Bench** (Wijk et al., 2024) are both learning engineering benchmarks, but differ in the complexity of the tasks and scenarios: MLE-Bench encompasses 75 tasks sourced from Kaggle[4], whose deterministic scoring functions are taken from the corresponding Kaggle competitions—as they score the final result, this allows agents to solve the task in ways potentially different from humans, going beyond substitution (Sec. 2.1); however, because these functions vary across tasks, each score is compared against a snapshot of the (human) leaderboard. Instead, RE-Bench includes 7 environments each presenting a unique Machine Learning (ML) task focused on optimising either the loss function or the run-time; the value of these scoring functions is manually inspected for evaluation, and evaluators need to have access to a reference solution. Relatedly, **MLAgentBench** (Huang et al., 2024a) comprises 13 tasks specified by a goal, occasionally constraints or specific instructions, starter files, and an evaluator; the tasks are collected and adapted from recent Kaggle challenges, CLRS (Veličković et al., 2022), BabyLM[5]; the starter files consist of data, description of data and metric, and initial code; each task has its own goal metric to improve on, whose measure is used for automated evaluation. For an overview of MLAgent workflow and evaluation, see Fig. 7. Instead, Nathani et al. (2025) introduce **MLGym**, an environment to train LLM agents on ML tasks using reinforcement learning. Given a task description, an initial codebase, and actions and observations history, the agent generates an action (shell commands executed by the environment) to accomplish research objectives iteratively; the execution feedback can then be used to refine the agent. MLGym is equipped with a benchmark consisting of 13 tasks spanning data science, game theory, computer vision, NLP, and reinforcement learning, and selected from sources such as Kaggle's House Price Prediction[6], 3-SAT (Cook, 1971), CIFAR-10's image classification[7], and more; they require the agent to perform Data Understanding, Modelling and Evaluation. As the various tasks have different performance metric, they score each agent by a quantity that reflects how closely, on average across a range of tolerance levels, it matches the best performer on every individual task. MLGym differs from MLAgentBench for the larger complexity of its tasks. Finally, (Li et al., 2025a) introduce **IDA-Bench**, which attempts to evaluate LLMs on their ability to perform guided predictive Modelling tasks; the benchmark includes an LLM-simulated user with domain knowledge and subjective insights who interacts with an agent to provide instructions throughout a multi-turn iterative process; the agent is then tested on adapting its goal and following instructions. The tasks are obtained from Kaggle; an LLM distils reference insights in natural language format from an optimal solution. These, together with information such as hyperparameters, serve as a task-specific template for the simulated user, which requests the agent to perform certain steps, without offering all insights up-front, and offers clarifications when asked. Results of the trained model are evaluated against a ground truth using task-specific evaluation functions; and compared with a human baseline, obtained by running the notebook the simulated user has access to. They also determine the ability to interact by considering how the prediction accuracy changes by increasing the number of interactions. Fig. 4 shows an example "trajectory" of the guided data analysis process.

Next, **CSR-Bench** (Xiao et al., 2025) and **SUPER** (Bogin et al., 2024) test whether agents can correctly deploy code from a project repository when given instructions—an important, though not exclusive, part of the data-science Deployment stage (Sec. 2.5). CSR-Bench and SUPER both transform GitHub repositories into end-to-end "run-the-code" challenges in which an autonomous LLM agent must parse documentation, install dependencies, debug failures, and produce a outcomes assessed by an automatic completion metric. CSR-Bench supplies 100 diverse repositories, each constituting one comprehensive task that typically involves environment setup, data and model acquisition, model training, inference, and evaluation. In contrast, SUPER targets reproducibility in machine-learning and NLP research across 801 repositories, organised into three nested subsets—*expert* (45 manually authored full-pipeline problems with human gold standards), *masked* (152 focused subtasks derived from the expert set), and *auto* (604 GPT-4-o-generated tasks created from repository `README`s)—each accompanied by task-specific metrics or expected outputs for evaluation.

Finally, considering agents performing Data Management tasks, **Spider 2.0** (Lei et al., 2024), the most recent iteration of Spider (Yu et al., 2018), is a benchmark of 632 real-world text-to-SQL workflow problems derived from enterprise-level database use cases; the agent's answers are evaluated using completion rate, accuracy, and coherence, therefore allowing agents to solve the task in ways different from pure human

---

[4]`https://www.kaggle.com/`

[5]`https://babylm.github.io/`

[6]`https://www.kaggle.com/datasets/zafarali27/house-price-prediction-dataset`

[7]`https://www.kaggle.com/code/faressayah/cifar-10-images-classification-using-cnns-88`

substitution (Sec. 2.1). Spider 2.0 differs from previous benchmarks by the same authors (Yu et al., 2018; 2019b;a, discussed in Sec. 3), in its more complex set-up: the tasks do not consist of pre-prepared inputs (question and database schema) or expected outputs (predicted SQL), but a real project codebase and a database interface; the agent interacts with the codebase through command scripts, as well as SQL queries.

## 4.2 Evaluating multiple activities explicitly

Some works target a broader spectrum of data science activities and evaluate each explicitly. To start with, **DA-Code** (Huang et al., 2024b), **InfiAgent-DABench** (Hu et al., 2024) and **DSBench** (Jing et al., 2024) all predominantly consider Data Preparation and Modelling, and mostly score the agent-produced solution by closely comparing it with reference ones, thus being anchored in the "substitution" paradigm (Sec. 2.1). In particular, DA-Code consists of 500 tasks sourced from Kaggle, GitHub, and other sources, each primarily covering exploratory data analysis (which roughly includes Data Understanding and Data Value Exploration), Data Preparation, or Modelling—thus, even though the overall benchmark consider multiple activities, each task is more narrow. DA-Code includes a variety of data structures and requires the use of SQL, Python, and Bash. Each task is accompanied by a single canonical artefact (table, chart, text file or hidden test-set labels) created by experienced annotators except for predictive modelling tasks. For grading, a solution is stripped down to the elements explicitly constrained by the instructions (such as required columns, the numeric data underlying a plot, or specified visual metadata) before applying a strict equality check against the reference artefact. For machine-learning tasks, the grader instead computes the task-specific metric (e.g. F1, MAE, Silhouette) on the hidden labels and awards partial credit in proportion to performance above baseline. Relatedly, InfiAgent-DABench introduces DAEval, a dataset of 257 GPT-4 generated closed-form questions, such as "Is there a linear relationship between the GDP per capita and the life expectancy score in Happiness_rank.csv? Conduct linear regression and use the resulting coefficient of determination (R-squared) to evaluate the model's goodness of fit ... [omitted for brevity]", derived from `csv` files sourced from GitHub repositories, with respective gold-standard answers generated by OpenAI's Advanced Data Analysis[8]. The benchmark covers a broad range of tasks, such as feature engineering, correlation analysis, data preprocessing, distribution analysis, summary statistics (all representing Data Preparation and Understanding), and machine learning (Modelling). The evaluation relies on calculating the portion of questions for which all subquestions exactly match the reference solution. Finally, DSBench obtains tasks from ModelOff[9] and Kaggle and split them into two categories: data analysis, 466 tasks characterised by long text context, various modalities, and a wide scope for solutions, and evaluated in terms of accuracy by an LLM which compares the responses to a human solution; and data modelling, 74 tasks requiring the LLM to build a predictive model with performance scored by the ability of the agent to generate and submit a bug-free model. Beyond Data Understanding, Data Preparation and Modelling, some tasks also cover Evaluation, Deployment, and exploratory activities.

Moving to a broader range of activities, **DSEval** (Zhang et al., 2024b) contains chains of inter-dependent problems (based on data from StackOverflow, Pandas-exercises[10], LeetCode[11], and Kaggle) where each highlights a different stage of the data-science lifecycle—Data Understanding and Preparation, Modelling, or interpretation (belonging to Deployment and Narrative Exploration)—while re-using the runtime context left by the previous problems. By doing so, agents must solve the overall task by following the same steps that humans would follow; thus, DSEval only evaluates agents' substitution ability rather than their potential to transform tasks (Sec. 2.1). For each problem, they employ custom validator modules to check correctness against the solution or run unit tests. Relatedly, **CoTa** (Li et al., 2025c) obtains a set of tasks by simulating (with LLMs) a company setup composed of an administrator and data scientist solving a client's problem making use of an AI Chatbot Agent; they then manually filter those interactions where the Chatbot Agent produced correct code and obtain 1024 interactions where agents are asked to write code to solve a problem or answer a multiple-choice question. Overall, these tasks cover Data Understanding and Preparation, Data Value Exploration, Modelling, Deployment, Results and Narrative Exploration (by converting plots into answers or summarising findings in prose). They test agents both in a "normal" mode,

---

[8] https://openai.com/blog/chatgpt-plugins#code-interpreter
[9] https://corporatefinanceinstitute.com/resources/financial-modeling/modeloff-guide/
[10] https://github.com/guipsamora/pandas_exercises
[11] https://leetcode.com/

where all requirements and details are specified by the user, and in an "action" mode, where the agent has to perform an action such as asking for clarification, updating code based on user-reported error, and others. For code updating and normal turns, the agent's interactivity is disabled, effectively falling back to an assistant setup, while in the other cases the agent may iteratively call a sandboxed Python executor. Further, for the "clarification" tasks, the agent has to pose follow-up questions that are answered by a LLM-simulated user, making this one of a few benchmarks (Li et al., 2025b;a) that evaluate interactivity. An example of the different interaction modes can be found in Fig. 5. Overall, however, scoring is purely based on the outcome (thus not judging the quality of the interaction): the agent's artefact is compared to a gold reference with task-specific comparators. Finally, **ScienceAgentBench** (Chen et al., 2024) builds on 102 tasks from scientific peer-reviewed publications, validated by subject experts; each task includes a data-driven discovery goal, information on the data, expert-provided knowledge, and a reference Python program. The questions are challenging, such as "Develop a drug-target interaction model with the DAVIS dataset to repurpose the antiviral drugs for COVID", or "Analyze Toronto fire stations and their service coverage to identify coverage gaps". The performance of the LLM agent on each task is scored against 3 metrics: Program Evaluation (itself consisting of: Valid Execution Rate, Success Rate, API Cost and embedding similarity computed by CodeBERT Zhou et al., 2023); Figure Evaluation (using GPT-4o); and Rubric-Based Evaluation based on 5 fundamental steps (Data Loading, Data Processing, Modelling or Visualisation, Output Formatting, and Output Saving). Therefore, this mostly evaluates substitution (Sec. 2.1) both by closely referring to human-provided solutions and by splitting the task in the same sequence of steps humans would follow.

### 4.3 End-to-end tasks scored by their final result

A few works instead evaluate agents on end-to-end questions—involving formulating plans, generating code and plots, and producing coherent results and insights—and score the final output of the task (in contrast to individual steps as in Sec. 4.2). This naturally allows to reward higher levels of task transformation beyond mere human substitution (Sec. 2.1). First, **InsightBench** (Sahu et al., 2025) includes 100 tabular datasets of 500 synthetically-generated entries each, organised in structures obtained from a real-world enterprise data management platform. When generating the synthetic data, a set of insights is manually "planted" in them. The insights (a total of 475) are divided into four families: descriptive, consisting of plots that describe the data; diagnostic, analysing the cause behind trends; predictive, consisting of visualisations that summarise model predictions; prescriptive, that explain actionable insights. The LLM agents are evaluated based on how many insights they recover, when provided with the dataset and an open-ended goal formulated by non-expert users, such as "Analyse incident trends in the data.csv file". In particular, Llama-3-Eval, a technique inspired by G-Eval (Liu et al., 2023b) which uses Llama-3 (Dubey et al., 2024), is used to compare the agent-produced insights with the reference ones, both at a summary level and at a deeper description level. The tasks require touch upon multiple data science activities—from Goal Exploration to Data Understanding, Value Exploration and Preparation, and to Modelling and Narrative Exploration—but only the final insights are directly evaluated (Deployment, Narrative Exploration, and Result Exploration). Similarly, **BLADE** (Gu et al., 2024) and **DiscoveryBench** (Majumder et al., 2024) both challenge LLM agents to explore a complex dataset with a vaguely defined goal, such as "Are soccer players with a dark skin tone more likely than those with a light skin tone to receive red cards from referees?" (Gu et al., 2024). However, differently from InsightBench, they consider scientific datasets and focus on agents' ability to integrate statistical knowledge with understanding of data from a broad range of scientific domains. Both BLADE and DiscoveryBench include end-to-end scientific data-analysis tasks that begin with genuine research questions and necessitate multistep solution workflows. BLADE includes 12 carefully curated datasets collected from statistical textbooks, research papers, and crowd-sourced studies, while DiscoveryBench offers 264 real-world tasks drawn from published studies plus 903 synthetic tasks spanning 48 domains. Tasks cover Data Understanding, Preparation, statistical and Machine Learning Modelling, Narrative Exploration, as well as Deployment, Data Value Exploration and various degrees of domain understanding (Business Understanding and Data Understanding). Both BLADE and DiscoveryBench grade solutions automatically with LLMs so that multiple defensible workflows can receive credit. Their emphases, however, diverge: BLADE checks if each analytical step—conceptual variable selection, admissible transformations, model family, hyper-parameters—corresponds to one of multiple expert-produced solutions (to account for alternatives), which however still limits the amount by which the agent can transform the task; DiscoveryBench

instead scores at the level of the final context, variables and relationship identified, using GPT-4 to judge the semantic match between the agent's claim and the human-produced reference solutions, without considering how the former was obtained. An overview of BLADE and DiscoveryBench can be found in Figs. 3 and 6. A recent paper (Hsu et al., 2025) has developed a benchmark, **WebDS**, to tackle the limitations of current benchmarks in evaluating LLM agents on their ability to browse and search the internet over multiple, multi-modal, unstructured websites to identify and collect datasets dynamically and perform the full trajectory of web data science activities, where they define a web data science task as one that requires navigation within an environment to acquire new raw information, that is then manipulated. They manually write 870 tasks based on 29 websites covering 10 data-heavy domains; they evaluate both the end results (using automated evaluation metrics where ground-truths are available), as well as the intermediate steps (such as subtask completion, tool usage, data validity, reasoning quality) using an LLM-as-a-judge which gives scores on a scale from 1 to 5, and then validating the protocol with human studies and stability analysis.

From Table 3, we can see that agents have been evaluated on more data science activities than assistants (Table 2), particularly considering goal-oriented activities (with the exception of Business Understanding); however, there is still a lack of evaluations for data management and, to a lesser extent, exploratory activities. Additionally, of all the surveyed works, only BiasBenchmark (Li et al., 2025b), CoTa (Li et al., 2025c), and IDA-Bench (Li et al., 2025a) evaluate agents in a collaborative framework with (simulated) users; whilst simulated users may be an oversimplification of real users interactions, and ignores the employment of domain experts in real-world scenarios, it is a step in the direction of evaluating interactions and their impact on performance.

## 5 Challenges and future directions

Our analysis shows that most evaluation works focus either on assistance, looking at isolated tasks that require LLMs to provide an answer on a single-turn basis (without access to tools and under human supervision) or on full automation, where LLMs are wrapped in agents that act autonomously. A few notable exceptions exist (Yu et al., 2019a; Li et al., 2025c;b;a), which primarily rely on other LLMs to simulate human users. While this approach ensures cost-effective evaluation and reproducibility, exploratory tasks may lead the agent to attempt novel solutions, such that the simulated user may be unable to assist it as the most suitable answer may not be within its knowledge base. This can be partly addressed by employing humans to answer such unprecedented queries and progressively enriching the knowledge base. Overall, however, the evaluation ecosystem should be enriched by more fully-fledged "centaur evaluations" (Haupt & Brynjolfsson, 2025) that directly evaluate human-AI collaborations, reorienting AI development towards augmentation rather than substitution and allowing researchers to directly measure human-centred desiderata, such as perceived helpfulness; among the surveyed works, only Li et al. (2025a) goes towards these more complex metrics, by attempting to quantify trade-off between autonomy and reliability/performance.

We found data management and exploratory activities remain mostly uncovered. This is due to 1) the inherent difficulty of scoring exploratory activities, which lack a fixed ground truth, and 2) the complex real-world interactions that certain exploratory activities (such as Business Understanding and Goal Exploration) and data management activities (Acquisition and Simulation) demand. To address this issue, simulated environments where data management and client interaction can occur should be developed, analogous to related developments in scientific research evaluation (Jansen et al., 2024; Cerrato et al., 2025). Such environments would enable the evaluation of agents or assistants that function holistically as data scientists by understanding business requirements, facilitating data collection, and adapting customer requests through data exploration. Simultaneously, realistic evaluation should progress toward end-to-end tasks that do not depend on strict ground truth comparisons or simple activity-specific metrics. Instead, these evaluations should reward insight generation (such as the works in Sec. 4.3) in potentially original ways, thereby properly incentivising systems that fundamentally redefine activities rather than focusing solely on human substitution.

Overall, these gaps in the evaluation landscape make it impossible to obtain a comprehensive characterisation of the performance of LLM-based systems across data science tasks and activities. This obscures potentially strong correlations between specific activities, which may better illuminate the underlying capability space and provide insights on how to best improve these systems going forward, following the traditional path in

machine learning where better evaluations in particular domains precede better systems. Thus, we propose the following actions to improve data science evaluation of AI systems:

- More comprehensive benchmarks covering most activities in Table 1, considering intermediate steps and preparatory activities, for evaluating substitution-focused approaches.

- Greater emphasis on incorporating human assistance (either real or simulated) in the evaluation, and developing methods to quantify the trade-off between autonomy and reliability.

- Development of comprehensive simulated environments that enable testing AI systems as holistic data scientists performing data collection and client interaction activities.

- Evaluations incorporating end-to-end tasks and broad objectives that allow and reward systems that redefine activities and propose original solutions differing from the reference ones.

- Field studies to validate the measurements obtained through these evaluation tools by comparing them to the real-world impact of human-AI collaborations.

- Once gaps in the evaluation landscape are filled, conduct correlation studies on performance across the different data science activities.

There has been enormous progress in data science automation, compared to the state of the art just a few years ago (De Bie et al., 2022). It is in open-ended tasks, the use of domain context and human-AI collaboration where data science automation is lagging behind, but upcoming tools may be able to conquer these domains: we must make sure our evaluations allow us to properly track progress.

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

# A  Appendix: definition of data-science activities from Martínez-Plumed et al. (2019)

We reproduce here for convenience the definition of the data-science activities as used in Martínez-Plumed et al. (2019).

The original stages of the CRISP-DM (Cross Industry Standard Process for Data Mining, Chapman, 2000) framework are as follows:

- **Business Understanding**: Understanding the project objectives and requirements from a business perspective, then converting this knowledge into a data mining problem definition and a preliminary plan designed to achieve the objectives.

- **Data Understanding**: Beginning with an initial data collection and proceeding with activities to familiarize oneself with the data, identify data quality problems, discover initial insights, and detect interesting subsets for hypothesis formation.

- **Data Preparation**: Encompassing all activities required to construct the final dataset from the initial raw data. This includes selecting tables, records, and attributes, as well as transforming and cleaning data for modelling.

- **Modelling**: Selecting and applying various modelling techniques while calibrating their parameters to optimal values. Some techniques have specific data requirements, often necessitating a return to the data preparation phase.

- **Evaluation**: Evaluating the constructed model(s) to ensure they properly achieve business objectives. The steps taken in the modelling process are reviewed to confirm that no important business issues have been overlooked.

- **Deployment**: Applying the model in a way that is useful for the customer, such as generating a report, implementing a repeatable data mining process, or integrating it into decision-making systems. While the customer typically executes deployment, the analyst ensures that all necessary steps are understood.

As argued in Martínez-Plumed et al. (2019), this framework assumes a well-defined business goal and pre-collected data. Additionally, it follows a fairly linear process, similar to mining metal in a known location. Thus, it is goal-oriented and process-centric, with data serving as an essential ingredient rather than the focal point. However, in exploratory data science, data takes centre stage, akin to prospecting rather than direct mining. Martínez-Plumed et al. (2019) introduces the following additional exploratory activities:

- **Goal Exploration**: Identifying business goals that can be achieved through data-driven approaches.

- **Data Source Exploration**: Discovering new and valuable data sources.

- **Data Value Exploration**: Assessing the potential value that can be extracted from the data.

- **Result Exploration**: Relating data science results to business goals.

- **Narrative Exploration**: Extracting meaningful stories, whether visual or textual, from data.

- **Product Exploration**: Identifying ways to transform extracted data value into services or applications that provide new and valuable benefits to users and customers.

Furthermore, Martínez-Plumed et al. (2019) critiques the CRISP-DM model for representing data as a static entity within the process, assuming that data has already been collected and merely needs understanding and preparation for modelling. However, modern data science projects often involve dynamic data management activities, including:

- **Data Acquisition**: Obtaining or generating relevant data, such as through the installation of sensors or applications.

- **Data Simulation**: Simulating complex systems to produce useful data and explore causal relationships (e.g., "what-if" scenarios).

- **Data Architecting**: Designing the logical and physical layout of data and integrating different data sources.

- **Data Release**: Making data accessible through databases, interfaces, and visualisations.

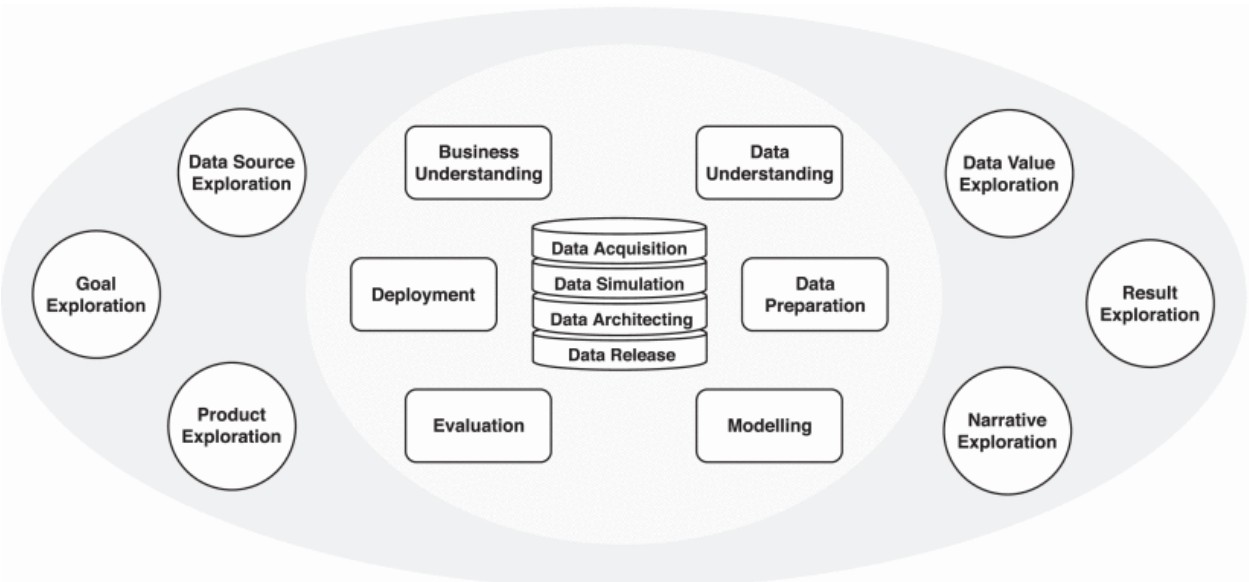

Figure 1: Martínez-Plumed et al. (2019): "The DST map, containing the outer circle of exploratory activities, inner circle of CRISP-DM (or goal-directed) activities, and at the core the data management activities."

## B   Appendix: examples of tasks

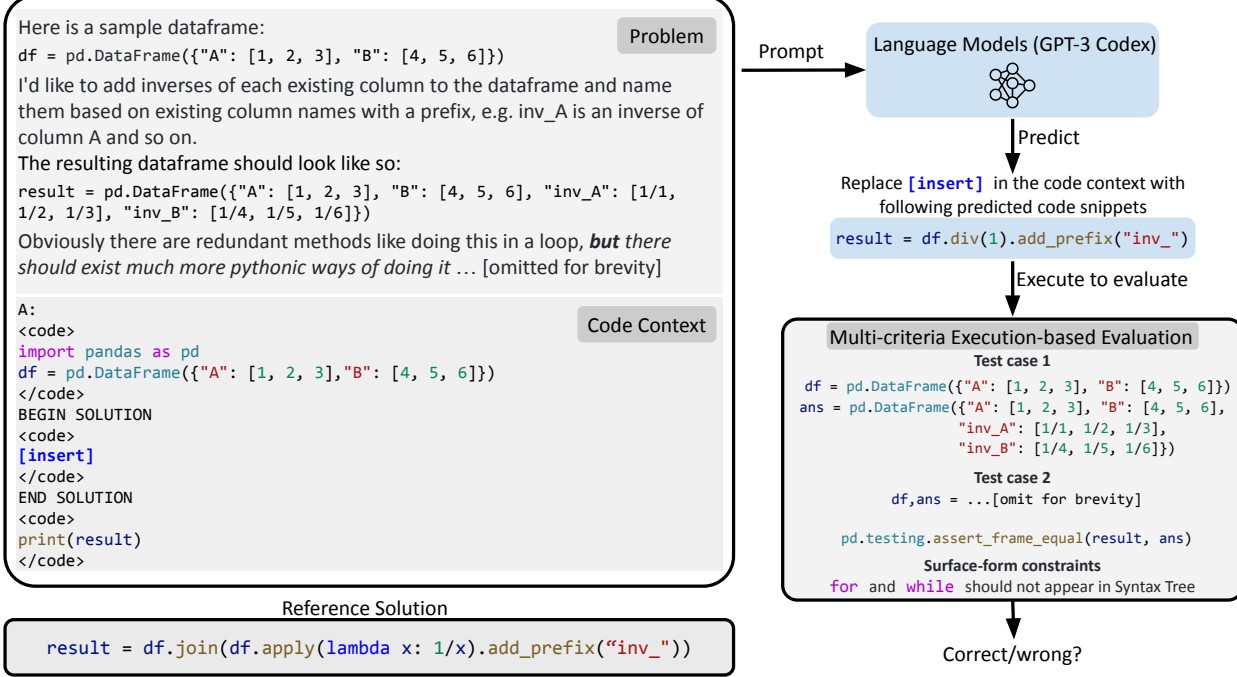

Figure 2: Lai et al. (2023): "An example problem in DS-1000. The model needs to fill in the code into [insert]in the prompt on the left; the code will then be executed to pass the multi-criteria automatic evaluation, which includes the test cases and the surface-form constraints; a reference solution is provided at the bottom left."

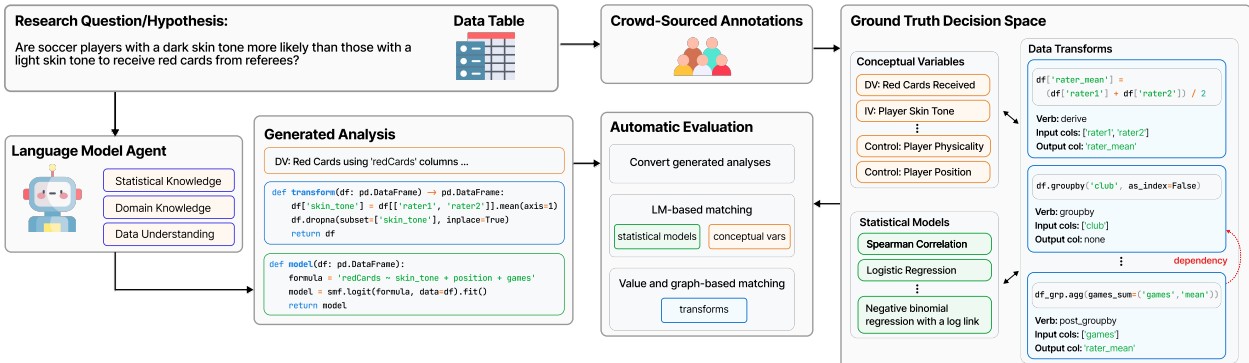

Figure 3: Gu et al. (2024): "Overview of BLADE. Given a research question and dataset, LM agents generate a full analysis containing the relevant conceptual variables, a data transform function, and a statistical modeling function (boxes 1-4-5). BLADE automatically evaluates this against the ground truth (box 6)."

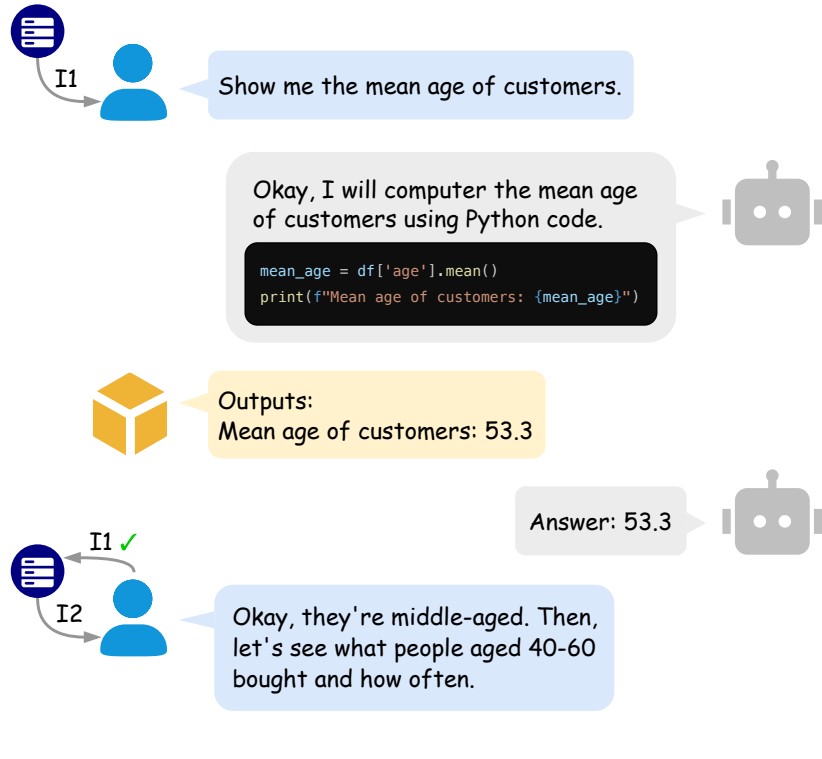

Figure 4: Li et al. (2025a): "Example task trajectory for Walmart sale prediction, showcasing the iterative interaction between the simulated user providing instructions and the agent executing code within the sandbox to achieve the analysis goal."

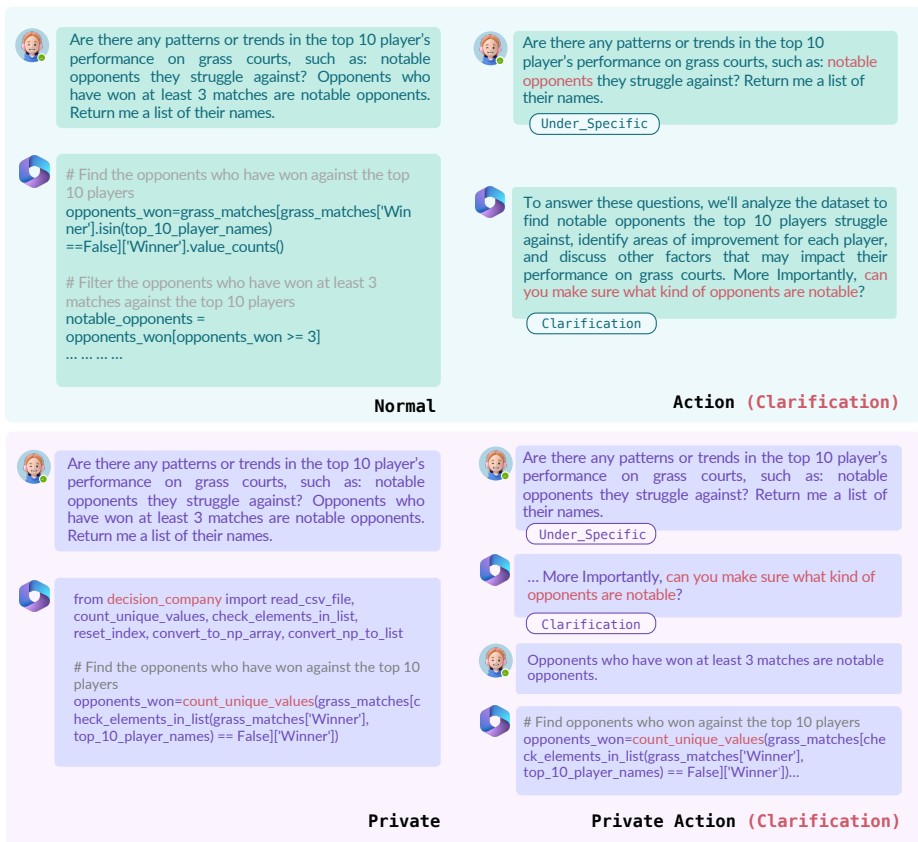

Figure 5: Li et al. (2025c): The figure shows the "Normal" mode, with the agent being provided all the relevant information and tasked with writing code to address the task, and "Action" mode, where the agent has to take a specific action (in this case, asking for clarification). "Private" refers to tasks requiring the use of bespoke software libraries to which the agent has access to.

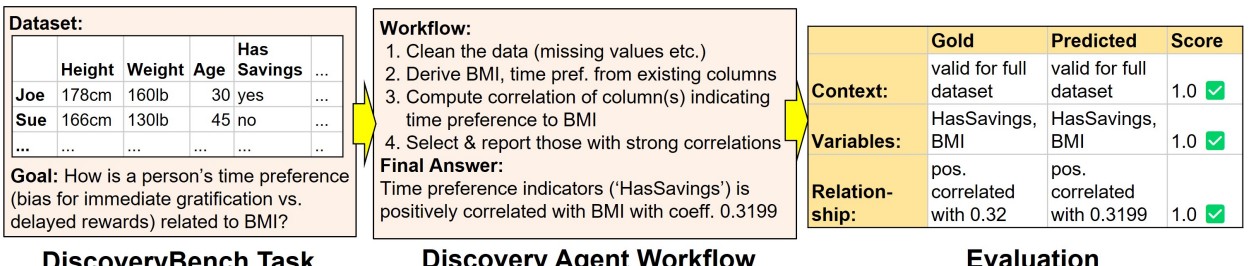

Figure 6: Majumder et al. (2024): "Each DISCOVERYBENCH task consists of a goal and dataset(s) (left). Solving the task requires both statistical analysis and scientific semantic reasoning, e.g., deciding which analysis is appropriate for the domain, and mapping goal terms to column names (center). A faceted evaluation allows open-ended final answers to be rigorously evaluated (right)."

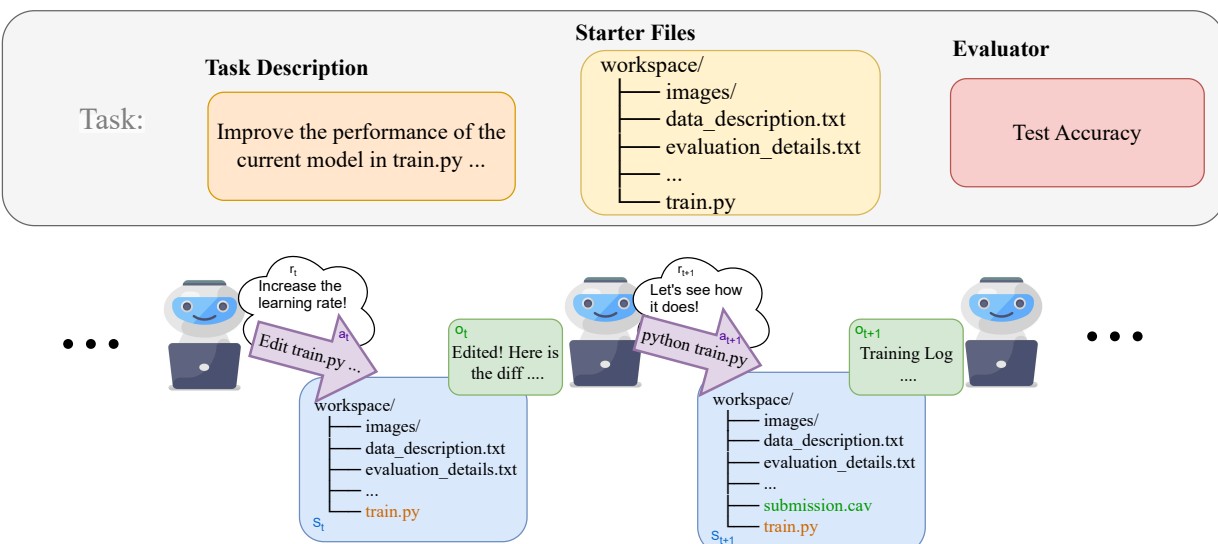

Figure 7: Huang et al. (2024a): "Overview of MLAgentBench. Each environment in MLAgentBench includes a task description, a set of starter files, and an evaluator. An agent can read/write files and execute Python code repeatedly, eventually producing a final file (e.g., test predictions in submission.csv). The agent is evaluated based on the quality of this file.."

