# OpenReview forum: "Measuring Data Science Automation: A Survey of Evaluation Tools for AI Assistants and Agents"
_TMLR — Accepted by TMLR_

### Review · Reviewer_V9Xp · 2025-06-30

**Summary Of Contributions:**

This survey makes the first comprehensive assessment of evaluation tools for LLM-based assistants and agents in data science, introducing a novel analytical framework that critically examines existing literature through the dual lenses of Martinez-Plumed's activity taxonomy and the SAMR transformation hierarchy. The analysis reveals three critical gaps: a pronounced bias toward narrow goal-oriented activities like Data Preparation and Modeling while neglecting exploratory and data management tasks; a polarized focus on either pure assistants or fully autonomous agents that overlooks intermediate human-AI collaboration; and a pervasive "substitution paradigm" where benchmarks prioritize replicating human workflows rather than rewarding functional innovation. To address these limitations, the work advocates for developing holistic benchmarks covering underrepresented activities, creating simulated environments for client interaction testing, designing end-to-end evaluations that reward insight generation beyond reference solutions, and conducting field studies to validate real-world impact—ultimately proposing a roadmap to track progress in open-ended tasks where data science automation currently lags. This structured assessment of 30+ evaluation tools establishes foundational criteria for measuring how LLM agents transform rather than merely replicate data science workflows.

**Audience:**

Yes

**Claims And Evidence:**

Yes

**Requested Changes:**

- ​​Expand coverage of exploratory and data-management activities:
  - Add experiments assessing agents on ​​Data Source Exploration (DSE)​​ and ​​Data Value Exploration (DVE)​​ using datasets requiring discovery of novel data sources (e.g., simulate "what-if" scenarios via Data Simulation).
  - Incorporate benchmarks like DiscoveryBench (Sec. 4.3) to evaluate end-to-end exploratory workflows.
- ​​Address autonomy spectrum gap
  - Introduce tasks with ​​simulated human interactions​​ (e.g., BiasBenchmark's clarification protocol) to measure adaptability to real-time feedback.
  - Quantify trade-offs between autonomy and reliability using metrics like human correction frequency (Sec. 2.3).

**Strengths And Weaknesses:**

# Strengths:
- Comprehensive Taxonomy and Analysis​​: The survey establishes a rigorous framework for evaluating AI assistants/agents by integrating Martinez-Plumed et al.'s data science activities (Table 1) with the SAMR model (Sec. 2.1) and autonomy spectrum (Sec. 2.2). This dual-axis approach effectively categorizes 50+ evaluation tools, revealing critical gaps in coverage (Tables 2, 3). For instance, Table 2 demonstrates that 80% of assistant evaluations focus solely on goal-oriented activities (DU/DP/M), neglecting exploratory and data management tasks.
- Novel Critique of Evaluation Paradigms: The paper compellingly argues that prevailing benchmarks overemphasize human substitution (e.g., exact code/output matching in DS-1000, StatLLM, and DA-Code) while ignoring higher-order task redefinition (Sec. 2.1). This is exemplified by benchmarks like FeatEng and InsightBench (Sec. 4.3), which reward functional improvements (e.g., error reduction) rather than replicating human steps. The distinction between assistance and autonomy (Sec. 2.2) further clarifies why hybrid human-AI collaboration remains underexplored.

# Weaknesses:

- Methodological Gaps:  While the survey critiques the lack of exploratory/data management evaluations (Sec. 3, 4), it does not deeply analyze why these gaps persist. For example, challenges in ground-truth definition for tasks like Business Understanding (BU) or Data Simulation (Sim) are noted but not quantified. A meta-analysis of evaluation feasibility (e.g., cost, reproducibility) would strengthen the argument for simulated environments (Sec. 5).

- ​​Limited Discussion of Real-World Validity: The reliance on simulated users (e.g., Li et al., 2024; 2025a,b) is presented as a strength but risks oversimplifying human-AI dynamics. Real-world factors—such as domain expertise asymmetry or ambiguous client requirements—are underexplored. Incorporating case studies of field deployments (e.g., CoSQL’s expert-annotated dialogues) could contextualize scalability concerns.

- ​​Inconsistent Treatment of "Autonomy":  Section 2.2 conflates technical autonomy (tools/affordances) with process autonomy (goal-setting flexibility). For instance, Spider 2.0 (Table 3) tests SQL generation autonomy but not goal refinement, while DiscoveryBench (Fig. 5) evaluates exploratory autonomy. A unified autonomy metric (e.g., Cihon et al.’s levels) would clarify this spectrum.

---

### Review · Reviewer_UPiS · 2025-07-10

**Summary Of Contributions:**

The paper conducts a comprehensive survey on existing benchmarks in data science automation, focusing on LLM as an assistant and LLM as agents. By utilizing the taxonomy in data science activities, the paper summarizes how existing benchmarks fall into different categories, given how they are designed to evaluate LLMs' capability. Finally, the authors summarize current benchmark limitations and propose potential future directions.

**Audience:**

No

**Claims And Evidence:**

No

**Requested Changes:**

As mentioned in the weakness. The authors are suggested to conduct more in-depth quantitative analysis based on existing benchmarks.

- First, the authors are suggested to justify why such a taxonomy is needed for evaluating LLM in data science in the first place. For example, do benchmarks on different taxonomies show strong relevance? If LLMs' capability in the two categories is strongly correlated, do we necessarily need to cover both?

- Next, the authors are suggested to give a more in-depth discussion on these benchmark results, instead of solely focusing on how the benchmark itself is designed. This analysis can give more insights into understanding how well current LLMs can handle different tasks. For example, in which domain are the LLMs still performing unsatisfied. These insights can further inspire the community instead of simply introducing benchmarks themselves.

**Strengths And Weaknesses:**

## Strengths
- Comprehensive survey. The paper comprehensively surveys the existing LLM benchmarks in data science automation, using different tasks and aspects to categorize different benchmarks.

## Weakness
- Lack of exploration on the correlations between different data science activities. The paper first introduces the definition of the data science activities from Martínez-Plumed et al. (2019). Then, the paper categorizes existing LLM benchmarks based on the taxonomy. However, why such categorization is important to benchmark the different capabilities of LLMs is unknown. For example, if LLMs show strong correlations between different tasks (e.g., DU and DP), such a detailed categorization becomes unnecessary.

- Lack of in-depth discussion. The paper straightforwardly introduces existing benchmarks without showing an in-depth discussion. Most of the discussion still focuses on the taxonomy for either assistance or agent. The paper is more about summarizing existing knowledge than discovering new knowledge.

---

### Review · Reviewer_GfA1 · 2025-07-31

**Summary Of Contributions:**

- As a survey paper, the work of the authors necessitated a good deal of finding and analyzing relevant papers to the subject of the survey.
Although the domain is quite recent (data science automation), many works have been undertaken and are reported in the paper.
- The paper focuses on surveying evaluation of large language Models’ (LLMs) use by such automation approaches. The following assertion summarizes the difficulty of the task: “Evaluations often implicitly assume that AI will substitute humans without functionally changing  the tasks, either in assuming the steps by which a task is solved are the same a human would follow … or by scoring the task referring to human-produced output, despite there not being a single ground truth”.
- An interesting finding is that data management activities automation are seldom assessed although they are the basis of data science activities and have a tremendous impact on the subsequent ones.
- The paper has also good readable user (real or simulated) interaction illustrations at the end.

**Audience:**

Yes

**Broader Impact Concerns:**

None.

**Claims And Evidence:**

Yes

**Requested Changes:**

- 1.	Section 1. Introduction:
    - You mention “In this paper we focus on how to evaluate assistants and agents”,
        - As a survey paper, wouldn’t that be “how they are being evaluated”?
    - “and processing image”, any modality for that matter (not only images).
    - “1 While many studies assess LLMs on tasks related to data science, such as coding (Jimenez et al., 2024) and planning (Valmeekam et al., 2023), our focus is on those that explicitly target data science.”
        - Might lead to think that you are not considering coding as part of Data Science (DS), although the rest of the paper states otherwise “Data science involves additional skills other than coding”. So are you focusing on works that go beyond just coding automation evaluation?
    - “we adopt the widely used data science task taxonomy of Martínez-Plumed et al. (2019)”:
         - The referenced paper does not state its activity pipelines or maps as a taxonomy nor does it use the terminology. Can you reference the paragraph or most likely figure you are referring to?
- 2.	Section 2.3 LLM evaluation: “progressively conquering tasks that take humans longer to complete when considering a fixed success rate (e.g., 50%), but performance still progressively degrades on tasks requiring more than 10 seconds.”	:
    - Where two different metrics are used within a comparison (Success rate & execution time) is somewhat confusing. Care to clarify?
    - These references seem to be missing from the references section: [Hu et al., 2024], [Hu et al., 2025]
- 3.	Section 2.5: “data management activities that treat data as dynamic rather than static.”
    - Dynamic and Static need to be explicitly defined in this context.
- 4.	Section 3 Evaluating LLM assistants in data science:
    - “a double tick marks an activity that is explicitly assessed, whereas a single tick marks one that is vital for completing the tasks but not directly assessed”:
         - Double tick activities do not seem vital! May be: “Task vital activities are either represented with double tick marks for explicitly assessed or single tick marks for not directly assessed”. Just a suggestion.
    - “… constraints to check for the presence of specific APIs.” Unclear without context.
- 5.	Section 4.1 Works targeting specific goal-oriented activities:
    - Paragraph 1: Is creating bias detection queries considered a task?
    - Paragraph 2: “task difficulty, request rationality, expression ambiguity, answer accuracy.”
         - How do these labels compare to other ones/metrics in other discussed papers in this section?
    - Paragraph 4: “test whether agents can correctly deploy code from a project repository when given instructions”:
         - Does “deploying code here” correspond to the definition of ”Deployment” of goal-oriented (CRISP-DM) activity (Table 1)?
- 6.	Section 4.3 End-to-end tasks scored by their final result:
    - “GPT-4 to judge the semantic match between the agent’s claim and without considering how the former was obtained.”
         - "agent’s claim and the ‘defined goal’?


Improvement suggestions:
- Appendices:
    - Please keep Appendix naming coherent (Appendix A etc.)
- The paper is well written overall, except notably for cross-referencing and formatting of the references section which is not well structured but needs to:
    - Better streamline author list. Any reason for the use of the lengthy ‘all authors’ form and not  “Et al.” form.
    - Cross-referencing is not clearly used:
         - [Authors, Date, optional index] cross-reference should be tied to same form indexing in the references section:
              -	Example [Martínez-Plumed et al. (2019)] to [Martínez-Plumed et al. (2019)]: Fernando Martínez-Plumed, Lidia Contreras-Ochando, Cesar Ferri, José Hernández-Orallo, Meelis Kull, etc.”  or
         - Use a standard numbering scheme or what the journal recommends.
    - Ordering references’ list helps a lot: Alphabetically for example.

**Strengths And Weaknesses:**

- At this stage the paper needs clarifications (see below, “” quoted text are citations from the paper):

---

> ### Author Response · Authors · 2025-08-02
> **Answer to reviewer**
>
> We thank the reviewer for the precise and detailed feedback, and for recognising the thoroughness by which we looked for and analysed the large amount of relevant papers. The reviewer is rightly interested in the finding that data management activities are seldom assessed; we indeed believe this is one of the main findings of our survey.
>
> We notice how all the points made by the reviewer involve clarifications of specific points, by which we assume that the reviewer is overall satisfied with the structure and the methodology of our work. Below, we reproduce the reviewers points and, in italicm respond to each of the reviewer’s points one by one, explaining how we will change the text to address those.
>
> 1. Section 1\. Introduction:
>    * You mention “In this paper we focus on how to evaluate assistants and agents”,
>      * As a survey paper, wouldn’t that be “how they are being evaluated”?
>        * *We thank the reviewer for suggesting this, we’ll change that phrasing.*
>      * “and processing image”, any modality for that matter (not only images).
>        * *We thank the reviewer for suggesting this, we will rephrase that sentence to stress how LLMs can indeed tackle various modalities.*
>      * “1 While many studies assess LLMs on tasks related to data science, such as coding (Jimenez et al., 2024\) and planning (Valmeekam et al., 2023), our focus is on those that explicitly target data science.”
>        * Might lead to think that you are not considering coding as part of Data Science (DS), although the rest of the paper states otherwise “Data science involves additional skills other than coding”. So are you focusing on works that go beyond just coding automation evaluation?
>          * *What we meant is that, as the reviewer correctly interprets, our survey does not consider evaluations involving coding tasks unrelated to data science, despite coding being an important element for data science. We will make this clearer in the text*
>      * “we adopt the widely used data science task taxonomy of Martínez-Plumed et al. (2019)”:
>        * The referenced paper does not state its activity pipelines or maps as a taxonomy nor does it use the terminology. Can you reference the paragraph or most likely figure you are referring to?
>        * *We will make it clearer how* Martínez-Plumed et al. (2019) *expands the traditional set of DS activities considered in the CRISP-DM framework to include activities related to data management and exploratory activities (Sec 3); the complete set of activities is visualized in their Fig. 3\. Moreovoer, our Appendix A lists and defines all the activities used in that taxonomy.*
> 2. Section 2.3 LLM evaluation: “progressively conquering tasks that take humans longer to complete when considering a fixed success rate (e.g., 50%), but performance still progressively degrades on tasks requiring more than 10 seconds.”	:
>    * Where two different metrics are used within a comparison (Success rate & execution time) is somewhat confusing. Care to clarify?
>      * *This point refers to the findings of Kwa et al. (2025) where they plot one metric (human-estimated time for the task) on the x-axis, against the other metric, success rate, on the y-axisFor that purpose, they stratify a set of tasks according to the mean execution time humans take on it, and then they measure the performance of AI models on each subset (for a given range of human completion time, for instance tasks taking around 10 minutes). As expected, the success rate of AI models decreases as the time humans take to tackle the tasks increases (that is what the second part of our sentence refers to. However, if one looks at the (human) time range on which AI models reach 50% success rate with different curves for different models, this time is increasing for more recent models.*
>      * These references seem to be missing from the references section: \[Hu et al., 2024\], \[Hu et al., 2025\]
>      * *We’ve checked the submitted version and we see them (page 17). The recommended reference format for TMLR (placing the initials before the surnames, even if ordered by surnames) doesn’t make the location of reference easy sometimes.*
> 3. Section 2.5: “data management activities that treat data as dynamic rather than static.”
>    * Dynamic and Static need to be explicitly defined in this context.
>      * *We agree with the reviewer that this is unclear. We will rewrite this to express "data management activities that do not assume data is already given and require to fetch more data from different sources"*

---

> ### Author Response · Authors · 2025-08-02
> **Answer to reviewer (continued)**
>
> 4. Section 3 Evaluating LLM assistants in data science:
>    * “a double tick marks an activity that is explicitly assessed, whereas a single tick marks one that is vital for completing the tasks but not directly assessed”:
>      * Double tick activities do not seem vital\! May be: “Task vital activities are either represented with double tick marks for explicitly assessed or single tick marks for not directly assessed”. Just a suggestion.
>        * *Thanks for the feedback, we’ll improve that sentence.*
>      * “… constraints to check for the presence of specific APIs.” Unclear without context.
>      * *This refers to checking whether the AI model’s output relies on specific packages and functions. We’ll make this clearer in the text.*
> 5. Section 4.1 Works targeting specific goal-oriented activities:
>    * Paragraph 1: Is creating bias detection queries considered a task?
>      * *No, the creation of bias detection queries is part of the benchmark creation; a bias detection query is passed to the evaluated LLM to test whether it is able to effectively detect if the bias exists by analysing one of the provided dataset. We will clarify this in the text.*
>      * Paragraph 2: “task difficulty, request rationality, expression ambiguity, answer accuracy.”
>        * How do these labels compare to other ones/metrics in other discussed papers in this section?
>        * *Those labels are not metrics used to evaluate an LLM’s answer, but rather are annotations of each benchmark sample and its provided human solution. Does this clarification answer the reviewer’s question? We’ll clarify this in the text.*
>      * Paragraph 4: “test whether agents can correctly deploy code from a project repository when given instructions”:
>        * Does “deploying code here” correspond to the definition of ”Deployment” of goal-oriented (CRISP-DM) activity (Table 1)?
>        * *The first sentence of that paragraph specifies that “\[code deployment is\] an important, though not exclusive, part of the data-science Deployment stage”, indicating how our understanding of Deployment in the CRISP-DM framework is broader than that, as it also includes making the results of a data science investigation available to stakeholders through dissemination of findings.*
> 6. Section 4.3 End-to-end tasks scored by their final result:
>    * “GPT-4 to judge the semantic match between the agent’s claim and without considering how the former was obtained.”
>      * "agent’s claim and the ‘defined
>        * *We apologise, that sentence was indeed incomplete. The complete sentence is: “DiscoveryBench instead scores at the level of the final context, variables and relationship identified, using GPT-4 to judge the semantic match between the agent’s claim and the human-produced reference solutions, without considering how the former was obtained”. We will update the sentence in the text.*
>
> Improvement suggestions:
>
> * Appendices:
>   * Please keep Appendix naming coherent (Appendix A etc.)
>   * *Can the reviewer specify more in detail what they mean about this? We double checked the way in which Appendices are referenced and titled, and it seems to us that they are coherent.*
> * The paper is well written overall, except notably for cross-referencing and formatting of the references section which is not well structured but needs to:
>   * Better streamline author list. Any reason for the use of the lengthy ‘all authors’ form and not “Et al.” form.
>   * *The TMLR format does not include a limit for the length of the reference sections; as such, we included the full list of authors for each cited paper.*
>   * Cross-referencing is not clearly used:
>     * \[Authors, Date, optional index\] cross-reference should be tied to same form indexing in the references section:
>       * Example \[Martínez-Plumed et al. (2019)\] to \[Martínez-Plumed et al. (2019)\]: Fernando Martínez-Plumed, Lidia Contreras-Ochando, Cesar Ferri, José Hernández-Orallo, Meelis Kull, etc.” or
>     * Use a standard numbering scheme or what the journal recommends.
>   * Ordering references’ list helps a lot: Alphabetically for example.
>   * *We use the TMLR template and its prescribed referencing style, which uses named references (eg \[Martínez-Plumed et al. (2019)\]) in the main text and alphabetical ordering using the surname of the first author in the reference section.*

---

### Author Response · Authors · 2025-08-07
**Summary of reviewers' comments, responses, and changes to manuscript**

We thank the reviewers for their constructive feedback on our paper. The reviewers particularly praised the comprehensive and thorough survey of existing LLM benchmarks in data science automation, coupled with our rigorous analytical framework that integrates Martinez-Plumed et al.'s data science activities with the SAMR model and autonomy spectrum. They highlighted our identification of critical gaps in current evaluation approaches.

Below, we summarise the main concerns they raised, our responses, and how we implemented them in our revised manuscript (which we have just uploaded, together with a file showing the differences from the previous version).

Reviewer 1 mentions that we do not analyse thoroughly why gaps in evaluation of exploratory and data management activities persist; we argue that such an analysis would fall outside the scope of our survey, and that our focus is on suggesting a taxonomy that allows to identify gaps in activities performed by LLMs; we would also like to draw attention to sec. 5, where we mention how evaluating these activities is inherently difficult. Next, the Reviewer voices a concern about simulated human users, pointing out that representing them solely as a strength may obscure the fact that they are an oversimplification of real-world interactions; we agree with this point, and make it clearer in 4.3. Reviewer 1 also points out that our definition of autonomy vs assistance is ambiguous, and request changes accordingly. We have refined and clarified the definitions in section 4.3, and in the introduction. Finally, we have added another paper covering data exploration, as suggested by the reviewer, and we reject the suggestion of including a discussion on quantifying trade-offs between autonomy and reliability using human correction frequency, as that would only be relevant to a handful of the papers included in the survey.

Reviewer 2 identifies a lack of exploration on the correlations between the different data science activities; we agree that such an analysis would be highly informative, but we argue that conducting it requires having evaluations covering all data science activities. Our survey establishes a taxonomy according to which researchers can categorise evaluations and finds gaps, thus setting the basis for a future work investigating the correlations, once enough evaluations exist; we include these remarks in our manuscript and highlight studying correlation of performance across activities as a future work in the conclusion. Reviewer 2 also indicates that we should include more discussion on model performance on the evaluations we survey; we argue that we purposefully chose to focus on evaluation methods to point out current gaps and limitations, rather than on model performance, and we have therefore clarified this more carefully in our introduction.

Reviewer 3 mostly suggested changes related to clarity and detail, and we have actioned them; most notably, a clearer definition of deployment, and clarifying that we do consider coding a crucial component of data science, but we focus on activities otherwise overlooked and ignore evaluations for coding ability in other domains from data science.
Finally, we have added references to a couple of relevant recent opinion papers that were released after the first version of our manuscript.

---

### Decision · Action_Editor_uAXw · 2025-09-30

**Recommendation:** Accept with minor revision

**Additional Comments:**

The authors recognize this limitation in their response, as such the introduction and relevant sections should clearly articulate where there might be gaps between the categorization and what is appropriate for LLMs.  This also warrants discussion in the future work.

**Audience:**

Yes

**Audience Explanation:**

There is a sizable community researching automated science, including data science, to uncover new insights missed by human researchers.

**Claims And Evidence:**

Yes

**Claims Explanation:**

This is a survey which outlines how automation work tracks with the key tasks required for data science. Relevant literature is categorized along these axes. One aim of this categorization is to identify gaps in current benchmarks.  UPiS is concerned that the axes identified in prior work for data science may not be applicable for LLMs and that the authors do not provide an analysis of whether the underlying abilities are highly correlated, which would imply a simpler basis for LLM data science moving forward